# Genome-Wide Association Study Meta-Analysis Elucidates Genetic Structure and Identifies Candidate Genes of Teat Number Traits in Pigs

**DOI:** 10.3390/ijms25010451

**Published:** 2023-12-29

**Authors:** Tingting Li, Pengchong Wan, Qing Lin, Chen Wei, Kaixuan Guo, Xiaojing Li, Yujin Lu, Zhe Zhang, Jiaqi Li

**Affiliations:** State Key Laboratory of Swine and Poultry Breeding Industry, National Engineering Research Center for Breeding Swine Industry, Guangdong Provincial Key Lab of Agro-Animal Genomics and Molecular Breeding, College of Animal Science, South China Agricultural University, Guangzhou 510642, China; litt197@163.com (T.L.); pengchong_wan@163.com (P.W.); qing_lin1996@126.com (Q.L.); weichenwjf@126.com (C.W.); kaixuan@stu.scau.edu.cn (K.G.); 18202053479@163.com (X.L.); lyjin@stu.scau.edu.cn (Y.L.); zhezhang@scau.edu.cn (Z.Z.)

**Keywords:** teat number, meta-GWAS analysis, cross-breed, PigGTEx

## Abstract

The teat number is a pivotal reproductive trait that significantly influences the survival rate of piglets. A meta-analysis is a robust instrument, enhancing the universality of research findings and improving statistical power by increasing the sample size. This study aimed to identify universal candidate genes associated with teat number traits using a genome-wide association study (GWAS) meta-analysis with three breeds. We identified 21 chromosome threshold significant single-nucleotide polymorphisms (SNPs) associated with five teat number traits in single-breed and cross-breed meta-GWAS analyses. Using a co-localization analysis of expression quantitative trait loci and GWAS loci, we detected four unique genes that were co-localized with cross-breed GWAS loci associated with teat number traits. Through a meta-analysis and integrative analysis, we identified more reliable candidate genes associated with multiple-breed teat number traits. Our research provides new information for exploring the genetic mechanism affecting pig teat number for breeding selection and improvement.

## 1. Introduction

Reproductive traits play a pivotal role in the pig industry, directly influencing the economic gains derived from domestic pigs [1]. Pig reproductive traits encompass various traits, such as teat number, litter size, and the birth weight of piglets. These traits collectively contribute to the overall reproductive performance and productivity of pigs [2]. The functional teat number in pigs serves as a critical indicator to assess sows’ lactation capacity and directly influences their offspring’s health and reproductive performance [3]. Teat number is characterized by low-to-medium heritability [4]. Traditional selection methods, such as phenotypic selection, are ineffective in improving these traits [5,6], which are primarily influenced by genetic factors [7]. These traits are determined at birth and are minimally influenced by environmental and nutritional factors [8]. Therefore, it is imperative to explore the genetic loci that affect teat number; this research direction has significant potential for uncovering the genetic mechanisms that govern teat number variation [9]. Although candidate genes related to the teat number traits have been identified through single-population GWAS (e.g., *VRTN* [9,10,11,12,13,14,15], *IGF2* [10,16], *SYNDIG1L* [12,14], and *LIN52* [17]), these genes are based on a single population with its genetic background and are not shared among multiple breeds. There are few reports on potential shared genes for teat number traits among different breeds.

A meta-analysis is a statistical method that combines results from multiple studies of the same type. Researchers use this method to summarize and statistically analyze data by pooling information across various studies. Meta-GWASs have been utilized to improve statistical power and mapping precision by combining single GWAS summary statistics. Zeng et al. [18] conducted a meta-GWAS on backfat thickness using 15,353 pigs to identify a shared genetic characterization of backfat thickness across breeds in pigs. Irene van den Berg et al. [19] performed a largescale meta-analysis in cattle to maximize both the power and precision of a GWAS for fat and protein percentages in milk. Higgins [20] conducted a GWAS followed by a meta-analysis in multiple breeds to identify genetic variants associated with residual feed intake and component traits (average daily gain and feed intake) in Irish beef cattle. A meta-analysis would facilitate a better understanding of the genetic factors influencing teat number across multiple pig populations [21]. Breeders are often attracted by the prospect of combining multiple populations and breeds as it enables the simultaneous application of the results to promote genetic improvement process across diverse breeds.

Different breeds each possess distinct features, and there exists a discrepancy in breeding goals. Specifically, the Duroc breed exhibits rapid growth, efficient feed utilization, and a high percentage of lean meat in the carcass, making it commonly used for terminal sires [22]. On the other hand, Landrace pigs are known for their excellent reproductive performance and have a high number of piglets per litter; this breed is commonly employed as the maternal line in commercial herds [23]. Meanwhile, Yorkshire pigs demonstrate strong stress resistance and resilience to rough feeding conditions, although their body sizes can vary significantly among individuals [24]. The breeding performance selection of Landrace makes its reproductive performance better than the other two commercial pig breeds, while Duroc’s growth performance is better than Yorkshire and Landrace. The growth and reproductive performance of distinct pig breeds exhibit significant variations, influenced not only by their genetic backgrounds but also by specific breeding objectives.

In this study, we used data from 5828 pigs across nine populations in China, encompassing three distinct pig breeds: Landrace, Yorkshire, and Duroc. The primary objective of the study was to identify significant SNPs and candidate genes associated with teat number traits, gaining deeper insights into the underlying genetic mechanisms governing teat number variation via an integrative analysis. This study contributes to expanding our genetic understanding of teat number, enabling advancements in optimizing reproductive performance and efficiency within pig breeding programs.

## 2. Results

### 2.1. Descriptive Statistics of Teat Number Traits and Genetic Background

The genotype data of nine populations are presented in Appendix A, which includes the number of test samples after carrying out quality control for the number of SNPs, chip types and reference genome versions. The descriptive statistics of the teat number traits are listed in Appendix A. In brief, there are 29,170 records of five teat number traits; the CV mean of TNUM_L (Left teat number), TNUM_R (Right teat number), TNUM_T (Total teat number), and TNUM_MPS (the maximum per side of teat number) were 6.78%, 7.22%, 5.78%, 7.11%, respectively, and for TNUM_D (the difference between sides of teat number), the CV mean is 186.11%. In individual-level data, we found that the heritability of teat number traits ranges from 0 to 0.44, most of the estimates are low-to-medium heritability (Appendix A), and genetic correlations of teat traits were strong and positive (Appendix A). Figure 1 shows the principal component analysis (PCA) results for combined populations, which indicate that the three breeds formed three distinct clusters. Individuals from different farms are close to each other within the same breed, and no outliers were observed. The first and second principal components account for more than 60% of the genetic variance.

### 2.2. Single-Population GWAS Identifies Genetic Loci Associated with the Single Population

To explore the SNP correlated with the teat number trait in each single population, we performed single-population GWASs for the five teat number traits in nine populations. For the chromosome threshold (1/N), we identified 66 SNPs that were correlated with five teat number traits via a single-population GWAS (Appendix A). Of these, we found 21 SNPs that surpassed the genome-wide significance threshold (0.05/N), and the result of the single-population GWAS is presented in Appendix A. Specifically, nine SNPs were associated with TNUM_L, and six SNPs were associated with TNUM_R, with four significant SNPs identified simultaneously on the Sus scrofa chromosome (SSC) 17. Furthermore, 11 SNPs were associated with TNUM_D, and 10 SNPs on SSC 8 were obtained from the F1 population. We discovered 22 SNPs related to TNUM_T, with the most significant SNP being 7_97614602 (*p* = 2.29 × 10^−7^). Additionally, 19 SNPs were associated with TNUM_MPS, and 4 of them were located on SSC 7.

In Duroc single-population GWAS results, 22 SNPs were related with teat number traits, and five SNPs were associated with TNUM_L, the most significant SNP on SSC 14 from the Duroc breed in the F2 population (14_3243635, *p* = 2.12 × 10^−11^). From Landrace pigs, 27 SNPs were identified; there are nine SNPs on SSC 7, 11 on SSC 8, and 14 SNPs related to TNUM_D. Among these SNPs, 10 SNPs are located on SSC 8. Regarding Yorkshire pigs, 17 SNPs were identified, and 9 SNPs were located on SSC 6 which were associated with TNUM_T and TNUM_MPS.

### 2.3. Single-Breed Meta-GWAS Analysis

To increase the power of the GWAS within the single breed, we performed a single-breed meta-GWAS in three different breeds. Figure 2 presents the meta-GWAS results for the Landrace breed; we identified 22 genome-wide significant SNPs beyond the chromosome significant threshold (Appendix A). Among them, three, four, six and nine SNPs were associated with TNUM_L, TNUM_R, TNUM_T, and TNUM_MPS, 7_96599577(three times); 7_97247184 (three times), 7_97614602 (four times) and 7_97440470 (two times) are repeated. Compared to the single-population results, 22 SNPs overlapped with the single population GWAS analysis, of which 8 SNPs were in SSC 7. After a meta-GWAS analysis, 10 SNPs had larger *p*-values. Yorkshire meta-analyses identified three significant SNPs located on SSC 13, 15, and 18 which were associated with TNUM_D, TNUM_T, and TNUM_MPS (Appendix A).

Duroc meta-analyses identified seven significant SNPs (Appendix A). Two SNPs on SSC 4 were related to TNUM_L and TNUM_T respectively, two significant SNPs on SSC 6 were associated with TNUM_T, and the remaining SNPs on SSC 1, 9, and 17 were related to TNUM_D and TNUM_MPS. Two SNPs coincided with the single population and were associated with TNUM_MPS and TNUM_R, and one SNP had a larger *p*-value after a single-breed meta-GWAS analysis.

### 2.4. Cross-Breed Meta-GWAS Analysis

By integrating cross-breed meta-analyses, we identified 14 significant SNPs (Appendix A); the results showed that two, one, four, and seven SNPs were correlated with TNUM_L, TNUM_R, TNUM_T, and TNUM_MPS, respectively (Figure 3), eight SNPs were consistent with the single-population results, and eight SNPs were consistent with the single-breed meta-GWAS analysis. It should be noted that SNP (7_97614602), located on SSC 7, appeared four times, affecting TNUM_L, TNUM_R, TNUM_T, and TNUM_MPS, and the minimum *p*-value of this SNP in the TNUM_MPS trait is 1.06 × 10^−12^; another SNP on SSC 7 (7_97440470) also affected two teat number traits (TNUM_T and TNUM_MPS), and an SNP on SSC 6 (6_166148395) appeared three times, simultaneously affecting three traits (TNUM_L, TNUM_T and TNUM_MPS). The cross-breed meta-GWAS identified four new SNPs (6_166148395, 3_128471318, 3_128412857, and 4_94047783) that were not detected in the single-population GWAS or single-breed meta-GWAS. Additionally, SNP 6_166148395 was found to be repeated three times.

### 2.5. Post-GWAS Analysis

After completing the GWAS analysis, we annotated the significant SNP expansions obtained upstream and downstream 1 Mb and generated 643 candidate genes. Subsequently, we performed gene ontology (GO) and Kyoto Encyclopedia of Genes and Genomes (KEGG) pathway analyses by integrating the results of the gene annotation, which facilitated the identification of biological pathways influenced by the genes under investigation. GO and KEGG enrichment (Appendix A) was performed to highlight the pathways and biological processes affecting teat number traits. Candidate genes for the teat number trait are mainly enriched in the extracellular exosome, nucleoplasm, and the integral component of plasma membrane. Nine genes were found to be enriched in the brain development pathway, with a *p*-value of 2.9 × 10^−2^. Annotated genes are mainly enriched in the metabolic pathway, thyroid hormone synthesis, and Cgmp-PKG signaling pathway.

For co-localization, the result shows a co-localization analysis of single-breed GWAS results, and eQTLs (expression quantitative trait loci) data have 2563 gene tissue–trait pairs and 507 significant pairs (PP4 > 0.85). We detected four genes that were significantly co-localized with the teat number traits. It should be noted that 2_18237854 and *ACCS* in the blastocyst (*p*_eQTL_ = 3.36 × 10^−5^), blood (*p*_eQTL_ = 3.11 × 10^−6^), muscle (*p*_eQTL_ = 1.44 × 10^−6^) and uterus (*p*_eQTL_ = 1.545257 × 10^−7^) had significant co-localization results and were was found to be related to TNUM_T and TNUM_MPS (Figure 4).

## 3. Discussion

In this study, we performed single-population GWAS analysis in 5828 pigs from nine populations and then conducted a single-breed meta-GWAS analysis to explore the genetic basis of teat number traits within each breed. This enabled us to detect the common SNPs that influenced teat number in each breed. Lastly, we carried out a cross-breed meta-GWAS to identify potential loci that were shared among the three breeds.

### 3.1. Meta-GWAS Results

The single-population GWAS has suffers from sample size insufficiency, and the single-population GWAS results are not obvious because of resource constraints, such as restrictions on labor and material and financial resources [25]. However, a meta-analysis is a quantitative approach that combines and scrutinizes results from similar studies. By pooling data across multiple investigations of the same type, a meta-analysis achieves the requisite sample size, thereby enhancing statistical efficacy and facilitating more reliable conclusions [16].

We compared single-breed meta-GWAS results with those of a single-population GWAS. The results showed that the single-breed meta-GWAS in Landrace pigs confirmed 17 significant SNPs detected in the single-population GWAS and newly identified four SNPs (4_31183347, 1_266058567, 1_266076710, and 4_31225507). The significant SNPs of Landrace were more than those of the other two breeds. In Duroc pigs, it confirmed five significant SNPs identified in the single-population GWAS and detected two additional SNPs (6_165972241, 6_166148395). However, in the Yorkshire pig populations, the meta-analysis only validated three SNPs detected by the single-population GWAS, and no new SNP was detected.

In China, the breeding plan for large-scale farms generally involves having an independent Landrace sow population in the breeding population which is crossbred with semen provided by Yorkshire boars in the core population to produce binary hybrid individuals. Then, the binary hybrid individuals in the breeding population are crossbred with Duroc boars in the core population to produce ternary hybrid individuals [14,26]. It is this breeding pattern that makes the breeding goals of pig breeds different, resulting in the outstanding reproductive performance of Landrace pigs [27]. For example, the Landrace teat number is greater than the other two breeds. In our results, the meta-GWAS results of the Landrace single breed also show a clear peak.

Significant SNPs with high reliability detected in a single-population GWAS may not be detected in a meta-analysis; the explanation for this situation is that the eligible SNPs in the meta-analysis may be in different linkage disequilibrium stages and have mutations in different populations, resulting in inconsistent allele directions and thus reducing the reliability of the combined-population GWAS analysis, even though it increased the sample size via a meta-analysis [11,13,28,29].

In our study, we detected two SNPs associated with teat number traits in the Landrace pigs using a single-population GWAS; these SNPs were in the QTL (quantitative trait loci) region previously reported to be associated with teat number traits on SSC 5. However, they were not confirmed by the results of cross-breed meta-analysis; from the cross-breed results, we discovered that the QTL region (SSC 7: 193.23–197.23 MB) associated with teat number has been reported 73 times in QTLdb [30], and the region of (SSC 7: 192.88–196.88 MB) has been reported 72 times; both regions were confirmed in our study to be related to teat number traits (Appendix A).

### 3.2. Genetic Parameters of Teat Number Traits

Many studies have estimated the genetic parameters of teat number traits in pigs, indicating that the teat number trait is a low- and medium-heritability reproductive trait. Several studies reported that the heritability range of TNUM_T is 0.09–0.36 [7,13,16,17], and the heritability ranges of TNUM_L and TNUM_R are relatively close, and the whole is in the range of 0.14–0.29 [13,17,18]. This is consistent with the heritability results of our single population calculation results, indicating that this trait is indeed heritable. In addition, the heritability of TNUM_D was the lowest among all traits, it should be noted that some traits have heritability values equal to 0. This could be due to insufficient population size and distant relative relationships, leading to inaccurate results [31].

We obtained results that closely matched the genetic correlation on the Venn diagram for genes annotated with significant SNPs for each trait. Many SNPs were related to each trait. In addition to TNUM_D, the trait-specific SNP also showed differences in influencing the genetic factors of each trait (Appendix A). The results of genetic correlation and phenotypic correlation can help us to learn the relationship and structure of traits and promote the breeding process through increasing understanding. According to genetic correlation, we can see that both phenotypic and genetic correlations show a high correlation, and the value of the correlation is close to 1, which is also consistent with our understanding that teat number may share a similar genetic regulatory structure.

For the significant SNPs associated with teat number traits, we discovered that some of them exhibited significant relationships with other traits. For instance, the SNP 7_102441757 demonstrated a strong association with loin muscle depth, and the genetic correlation between teat number and loin muscle depth was determined to be 0.99 from PigBiobank [32]. Additionally, teat number and backfat thickness displayed noteworthy loci, such as 7_97614602, with a genetic correlation of −0.3068. This indicated that these SNPs possess the potential to influence a multitude of intricate traits [22]. Consequently, these SNPs warrant further scrutiny in the context of breeding programs. This is due to their potential role in trait variation, which could be harnessed to enhance desirable traits in breeding populations. These findings provide further insights into the potential genetic mechanisms underlying the interplay between different traits.

### 3.3. Candidate Gene

In this study, we validated several potential genes identified in previous studies and discovered novel genes that may influence traits. Previous studies identified *VRTN*, *SYNDIG1L,* and *NUMB* as the primary genes with the potential to influence teat number traits. For *VRTN*, it was the most convincing gene affecting teat number by GWAS, and the *VRTN* gene mutation was significantly associated with the thoracic vertebra number, which influenced teat number potentially [9,17,21,33,34,35,36,37]. The *SYNDIG1L* gene is located on SSC 7, and it was shown that *SYNDIG1L* acted as an annotation gene near the SNP that most affected the teat number trait [14,34,38]. The genes related to the number of vertebrae, such as *VRTN*, and *SYNDIG1L*, may be related to the teat number traits in pigs [39]. The *NUMB* gene is the endocytic adaptor protein, and the protein encoded by this gene plays a role in the determination of cell fates during development. The degradation of the encoded protein is induced in a proteasome-dependent manner [40]. The *VRTN* gene is located near the significant SNP (7_97614602) on SSC 7, and it simultaneously appeared in the single-population GWAS results, Landrace meta-analysis, and cross-breed meta-analysis; in the PigGTEx resource, it is highly expressed in the blastocyst compared to other tissues.

*U6* was found to be associated with five traits simultaneously and was identified as the most likely potential gene linked to the trait in our study. The *U6* gene is a small nuclear RNA (snRNA) component of the *U6* snRNP (small nuclear ribonucleoprotein), which is highly conserved across species and is deemed to be important in splicing activity [41]; its role is to bring the catalytic site to the splice site. *LTBP2* is a protein-coding gene that plays an integral structural role [42]. The gene under investigation is associated with the number of ribs in a Large White × Minzhu intercross pig population. Additionally, it has been observed to be linked with the number of thoracic vertebrae in the same population [43,44]. In our study, we found it to be correlated with four teat number traits (TNUM_L, TNUM_R, TNUM_T, and TNUM_MPS); it is considered a candidate gene that may influence teat number traits. And we found it was the significant eQTL for 7_97614602 (*p*-value = 1.56 × 10^−4^). We detected nine genes that were significantly co-localized with the teat number traits in different tissues; however, our GWAS data were based on chip data, while eQTL data are based on PigGTEx, which has more genetic loci, and there is a discrepancy between the two sets of data.

In our study, we conducted a single-population GWAS, a single-breed meta-GWAS, and a cross-breed meta-GWAS on 29,170 teat number trait data records from 5828 pigs. We detected a total of 60 SNPs, out of which 12 were detected only in the meta-analysis. By annotating the meta results, we obtained candidate genes for further analysis. *VRTN*, *SYNDIG1L,* and *NUMB* genes were verified, and *U6* and *ACCS* genes were newly found to be related to the teat number traits. We enhanced the power to elucidate the regulatory mechanism of complex traits using an integrative analysis. The results of this study can help us understand the genetic similarity between the same breed and the heterogeneity between different breeds and provide a new idea for exploring the genetic mechanism affecting pig teat number for variety selection and improvement.

## 4. Materials and Methods

### 4.1. Ethics Approval and Consent to Participate

These considerations were not applicable since no biological samples were collected and no animal handling was performed for this study. All data were collected from existing databases provided by South China Agricultural University (Guangzhou 510642, China).

### 4.2. Phenotypes

The phenotype data for teat number traits were collected from 5828 pigs when they reached approximately 100 kg (100 ± 5 kg) body weight and 150 days (150 ± 5 days) of age. The data were recorded from four breeding farms (F1, F2, F3, and F4), including three breeds (Landrace [N = 1459], Yorkshire [N = 1853], and Duroc [N = 2516]); among them, there were 2030 boars and 3768 sows. In our analysis, we examined five teat number traits, including four quantitative traits (left teat number [TNUM_L], right teat number [TNUM_R], total teat number [TNUM_T], and the maximum per side of teat number [TNUM_MPS]); one derived trait was calculated as the difference between the number of teats on each side (TNUM_D).

### 4.3. Genotype Data

The pig genotyping was performed using two kinds of SNP chip, a “ZhongxinI” 50 K Porcine Breeding Chip (Beijing Compass Agritechnology Co., Ltd., Beijing, China) and a GeneSeek GGP-Porcine Beadchip (Neogen Corporation, Lansing, MI, USA) with 50 K. The genotype data quality control criteria were as follows, and quality control was performed using Plink v1.90 [45] software: (1) the sample call rate greater than 0.90 was retained; (2) the SNP call rate higher than 0.90; (3) the minor allele frequency (MAF) higher than 0.01; and the (4) *p*-value < 10^−6^ for the Hardy–Weinberg equilibrium test was excluded. After quality control, the number of SNPs within each population is shown in Appendix A.

As the animals in this study came from different farms; to identify and correct population stratification among the test animals, a principal component analysis (PCA) was carried out via a genome-wide complex trait analysis (GCTA) v1.94 [46]. Principal components were also used as covariates for subsequent analyses.

### 4.4. Estimation of Genetic Parameters

The narrow-sense genetic parameters of teat number traits, including heritability and genetic correlation, were estimated using GCTA v1.94 software in single populations. The statistical model for the estimation of heritability is shown below:(1)y=Xβ+Wα+e.
where y represents the vector of the phenotypic values of teat number traits; *β* represents fixed effects including sex and the first five principal components; *α* represents additive genetic effect vectors for individuals, *a*~N(0, G$\sigma_{a}^{2}$); $\sigma_{a}^{2}$ is the additive genetic variance, *G* is the genomic relationship matrix; *X* and *W* are incidence matrices for *β* and *α*, respectively; and *e* is the vector of residual with *e*~N(0, $I\sigma_{e}^{2}$), where $\sigma_{e}^{2}$ is the residual variance and I is the identify matrix. We used sex and the first five principal components as covariables for five teat trait genetic correlation estimates in the bivariate model.

### 4.5. Single-Population GWAS

The single-population GWAS analysis was conducted for each population using the fastGWA [47] module of GCTA v1.94, and the linear mixed model of the GWAS is shown below:(2)y=Xβ+Wb+Zg+e.
where y is the vector of teat number phenotypes; *X* is a vector of marker genotypes of the variant; and *β* is the effect of the variant. *W* is the incident matrix for the fixed and covariates, containing sex, and the top five principal components (PCs), and *b* is the corresponding effects. *Z* is the design matrix for the additive effect. $g\sim (0,\;G\sigma_{g}^{2}$), *g* is the genetic effect, $\sigma_{g}^{2}$ is the additive genetic variance, and G is the sparse genetic relationship matrix (GRM). $e\sim (0,\;I\sigma_{e}^{2})$, *e* is the residual effect, $\sigma_{e}^{2}$ is the residual variance, and *I* is the identical matrix.

The genome-wide and suggestive significant thresholds for the single-population GWAS results were set at 0.05/N and 1/N, respectively, 0.05/N represents the significant level of the whole genome, and 1/N represents the significant level of the chromosome, where N represents the number of SNPs used for analysis in the study.

### 4.6. Meta-GWAS Analysis

The following two strategies were applied to the meta-GWAS analysis:(1)A single-breed meta-GWAS analysis combined with same-breed GWAS summary results; nine populations were divided into three breeds for a meta-GWAS analysis.(2)A cross-breed meta-GWAS analysis.

The cross-breed meta-GWAS analysis was performed between three breeds by combining the GWAS results for single populations and all breeds by combining the GWAS results of various breeds. The above two strategies were both completed using METAL software (released on 2020-05-05) [48] and by calculating the pooled inverse-variance-weighted β-coefficients, standard errors, and Z-scores, and the formulas were as follows:(3)$$w_{i}=1/{SE}_{i}^{2}$$
(4)$$se=\sqrt{1/\sum_{i}w_{i}}$$
(5)$$\beta =\sum_{i}{\beta_{i}w}_{i}/\sum_{i}w_{i}$$
(6)z=β/SE
where *β_i_* is the β-coefficient for study *i*; and *SE* corresponds to the standard error for study *i*.

### 4.7. Functional Annotation of Candidate Genes

All SNP physical locations are based on the Sus scrofa genome (version 10.2), and we extracted the 1 MB upstream and downstream genes of significant SNP sites in meta-GWAS, annotated functional genes using the NCBI (http://www.ncbi.nlm.nlh.gov/, accessed on 15 May 2023) and Ensembl databases (http://www.ensembl.org/, accessed on 15 May 2023), and found candidate genes related to teat number traits. In addition, David (http://david.ncifcrf.gov/tools.jsp, accessed on 15 May 2023) was used to conduct an enrichment analysis of the annotated genes, including a gene ontology (GO) [49] analysis and a Kyoto Encyclopedia of Genes and Genomes (KEGG) [50] pathway analysis. The statistical method used in the enrichment analyses was the Fisher statistical test, and the results showed enrichment of GO entries or the KEGG pathway with the *p*-value < 0.05. The results of the GO and KEGG plots were shown using R software v4.1.2 [51].

### 4.8. Co-Localization Analysis

To explore the potential regulatory mechanism of the gene affecting the teat number traits, we performed a co-localization analysis using the coloc R package [52] based on 1 Mb up- and downstream of the most significant SNPs from the GWAS summary and the eQTL data from PigGTEx (http://piggtex.farmgtex.org/, accessed on 6 August 2023). Finally, we set the posterior probability 4 (PP4) larger than 0.85 as the significant threshold to filter candidate genes.

## Figures and Tables

**Figure 1 ijms-25-00451-f001:**
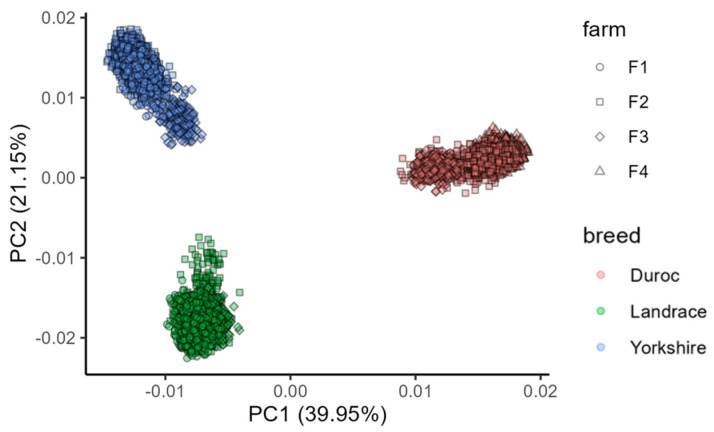
Principal component analysis of the combined population.

**Figure 2 ijms-25-00451-f002:**
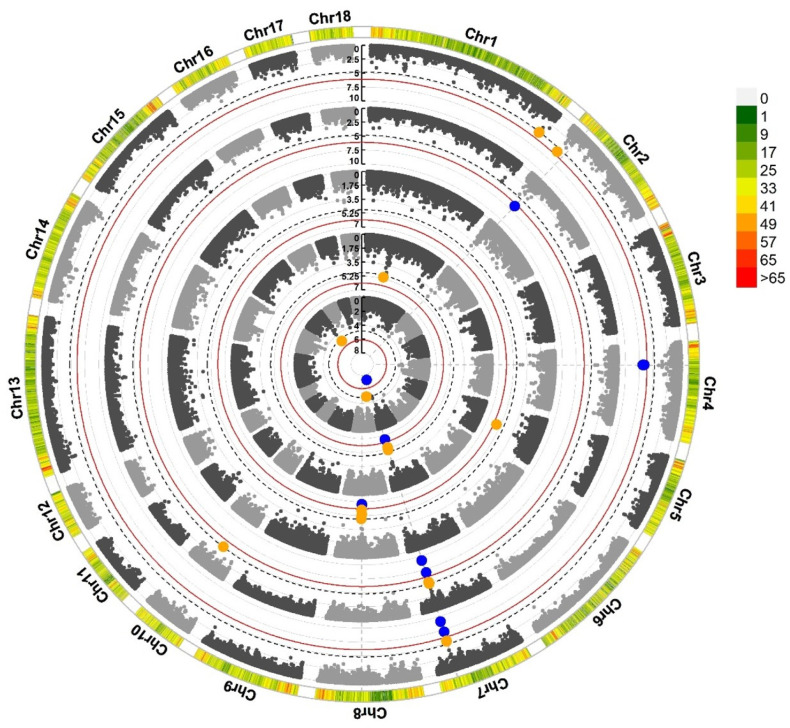
Circular Manhattan of Landrace pig meta-analysis results. The blue dots represent SNPs whose *p*-values exceed the genomic significance threshold; the orange dots represent SNPs whose *p*-values exceed the threshold of chromosomal significance. The outermost circle of the figure represents the density of SNPs. The five phenotypes from inside to outside are TNUM_L, TNUM_R, TNUM_D, TNUM_T, and TNUM_MPS, respectively.

**Figure 3 ijms-25-00451-f003:**
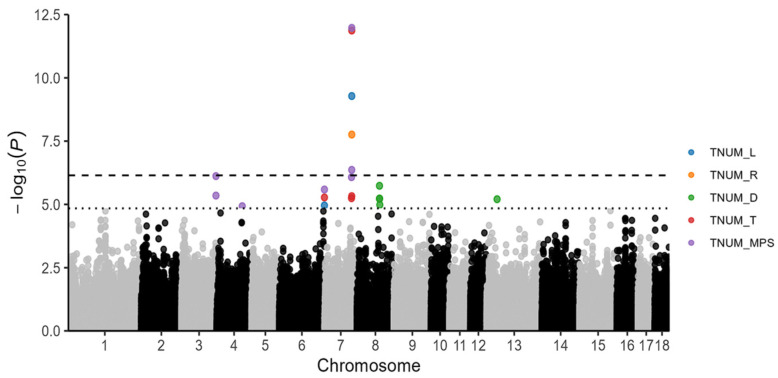
Manhattan of meta-analysis results for cross-breed; different colors represent different teat number traits. The black and gray colors represent variants on different chromosomes that are below the significance threshold.

**Figure 4 ijms-25-00451-f004:**
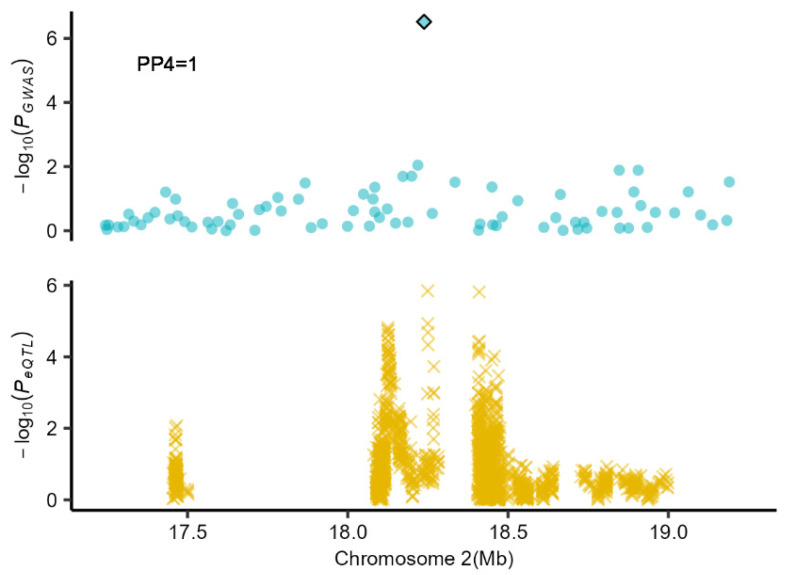
The local Manhattan plot for the independent variants 2_18237854 and eQTL mapping in the single-breed GWAS meta-analysis (muscle). The top panel was the meta-GWAS analysis result, and the blue rhombus was the independent variant 2_18237854, the bottom panel represents the eQTL mapping results in muscle.

## Data Availability

The datasets analyzed during this study are available from the authors upon reasonable request.

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
