# Peer review of "Genome-Wide Association Study Meta-Analysis Elucidates Genetic Structure and Identifies Candidate Genes of Teat Number Traits in Pigs"

_ijms, 2023, doi:10.3390/ijms25010451_

Round 1
Reviewer 1 Report
Comments and Suggestions for Authors
The manuscript presents valuable results on the genetics associated with the number of teats in sows. Since this reproductive trait is important also from economical point of view, the findings of the study will help in further selection and breed development for pig industry. The overall quality of the paper is very high. Specifically, the Introduction section might be reformed as the first two paragraphs can be synthesized, so that the section describing the importance of teats in sows become more concise.
The material and methods section is described in sufficient details.
The results are clearly presented. A specific remark concerns 2.3. Single-breed meta-GWAS analysis with the single breed. This subtitle should be corrected to 2.3. Single-breed meta-GWAS analysis.
The graphic material of the manuscript is adequate.
Concerning discussion, it might be seen that for some part of the results (Candidate genes) there are many references cited. For the rest of the subsection, there is hardly, 1 or 2 references cited. Are the studies in the are so scarce?
The conclusions are supported by the data and are well formulated revealing future prospects for studies concerning the genetics of this important trait.
Reviewer 2 Report
Comments and Suggestions for Authors
Dear Authors,
The presented manuscript touches on important reproduction traits that result in higher economic benefits. The study included a huge pig population from four farms, including three breeds: Yorkshire, Duroc and Landrace. I see that the manuscript was previously submitted, and the present version was thoroughly corrected compared to the past version. I suggest only minor corrections that can improve or enrich the manuscript for the readers. Overall, I found numerous typing errors in starting sentences with low letters. Please check.
Introduction
In my opinion, this section lacks information about candidate genes associated with teat number, and what is known about this theme. I know that in the discussion section, it was discussed candidate genes linked to teat number but in the context of obtained results. In the introduction section, it should be given, examples of these candidate genes for teat number, which do not necessarily correspond with obtained results, but they described in the literature.
Moreover, in this section it should be added some information about the pig breeds used in the present study. Are they used as maternal or paternal components in the breeding? Which breeds are the most important in the context of teat number? For example in our country landrace is used as a maternal component in breeding and duroc pigs as paternal, so an improvement of teat number in the duroc pigs is not so important as in landrace. This information it should be given in the context Chinese breeding. Additionally, how the mean number of teats in the used pig breeds?
Discussion
please indicate which breeds are used as maternal or paternal components in Chinese breeding.
line 213 if less affected by genetic factors? so what can affect this trait, the environment?
methods
in the investigation were used only gilts? or also sires?
